# Obstacles to university food pantry use and student-suggested solutions: A qualitative study

Aseel El Zein[1][¤], Melissa J. Vilaro[2], Karla P. Shelnutt[2], Kim Walsh-Childers[3], Anne E. Mathews[1]*

1 Food Science and Human Nutrition Department, University of Florida, Gainesville, Florida, United States of America, 2 Department of Family, Youth, and Community Sciences, University of Florida, Gainesville, Florida, United States of America, 3 Department of Journalism, University of Florida, Gainesville, Florida, United States of America

¤ Current address: Department of Nutrition Sciences, University of Alabama at Birmingham, Birmingham, Alabama, United States of America

* anne.mathews@ufl.edu

**Data Availability Statement:** Data cannot be shared publicly because of confidentiality and privacy concerns. The University of Florida Institutional Review Board approved the study, and

## Abstract

### Background

In the absence of federal programs and policies to alleviate college student food insecurity, the number of food pantries has grown rapidly in the United States. Yet, no studies, to date, have qualitatively examined students' experiences with this resource.

### Objective

To explore college students' perspectives on barriers to using an on-campus food pantry and provide insights into student-suggested solutions.

### Methods

In this qualitative study, 41 college students were recruited from a large public university in the southeastern US with a campus food pantry. Students participated in one-on-one, in-person, semi-structured interviews. All interviews were audio-recorded, transcribed verbatim, managed using NVivo 12, and analyzed using inductive, semantic thematic analysis.

### Results

Most students were classified as food insecure (n = 33, 82.5%), and two-thirds identified as pantry users (at least once). The students' reasons for not using the food pantry indicated resistance and access barriers. Students either 'chose not to use' the campus food pantry due to (i) stigma and shame, (ii) perceived insufficient need, (iii) and unsuitable food or they experienced 'barriers' due to (i) lack of knowledge and (ii) limited food access. The main reason reported by food insecure non-pantry users was feelings of stigma and shame while that of food insecure pantry users was limited food access. Students suggested three solutions to minimize barriers experienced when utilizing the campus food pantry. These

the consent document the IRB approved assured participants that their data would not be shared beyond the research team and as aggregated in publications. To request data underlying this manuscript, please contact Dr. Anne Mathews at anne.mathews@ufl.edu. You may also contact the University of Florida Institutional Review Board for any questions at 1-352-392-0433 (USA) and reference IRB201801405.

**Funding:** UF STEM Translational Communication Center and UF Clinical and Translational Science Institute, awarded to AM. The funders had no role in study design, data collection and analysis, decision to publish, or preparation of the manuscript.

**Competing interests:** The authors have declared that no competing interests exist.

included (i) spreading awareness about the pantry through positive marketing messages that de-stigmatize use, (ii) improving accessibility of fresh produce and protein options, and (iii) improving access through satellite locations and online ordering systems.

## Conclusion

These barriers need to be systematically addressed to normalize food pantry use. Consideration of student recommendations by university program developers and policymakers may be of added value to expand access to food by college students with food insecurity.

## Introduction

Food insecurity is a substantial problem among college students in the United States (U.S.) with estimates suggesting that as many as half of American undergraduates struggle with having consistent access to food [1]. Defined by the U.S. Department of Agriculture as the "limited or uncertain availability of nutritionally adequate and safe foods, or the inability to acquire foods in socially acceptable ways" [2], food insecurity is associated with poor academic performance [3], lower rates of degree completion [4], and reduced mental and physical health [5]. In the U.S., the Supplemental Nutrition Assistance Program (SNAP) provides the nation's largest safety net to combat food insecurity; however, college students face barriers to accessing federal food aid through SNAP [6]. In 1980, federal law prohibited college students who are enrolled at least half-time from receiving SNAP benefits [6]. While the federal law established some exemptions, these are designed to narrowly target students in need of assistance. For example, in addition to all the other SNAP eligibility criteria, full-time students are eligible for SNAP if they: are committed to working at least 20 hours a week, have dependents between the ages of 6–11 and have no childcare, have a disability, participate in work-study programs, or have other waivers [7]. Limited access to SNAP benefits is based on the notion that college students receive financial support from families and schools. However, many college students are now considered "non-traditional" [8], described as having a single parent status, being heads of households, having a full-time job, being financially independent from their families, and often rely on their limited income to support housing, living, and educational expenses.

As a response to these limitations, non-profit food assistance programs, like on-campus food pantries, have grown rapidly to more than 800 nationwide, providing a short-term relief to those who choose to access this resource [9]. Despite this increase in the number of food pantries, findings from preliminary reports suggest that a small fraction of food insecure students utilize this resource for food acquisition [10]. Based on a sample of 899 students at the University of Florida, only 38% of students with food insecurity utilized the pantry although 66% were aware of its existence [10]. Some previously reported barriers to food assistance from non-student samples include the perception that relying on such resources violates one's ideals of self-sufficiency and exposes users to a high stigma service. Interestingly, interviews conducted by Kissane [11] revealed that the amount of stigma varies depending on the type of social service offered. For example, an after-school food program had lower perceived social stigma compared to using a highly visible food bank [11]. With the current normalization of the "starving student college lifestyle" [12] and reports of underutilization of campus food assistance [10], questions remain whether students share similar perspectives on food assistance. Understanding the reasons students would decline nonprofit assistance despite financial need is important to target individuals who may benefit from such resources.

To date, no study has used thorough qualitative methods to gain deeper insights into students' perspectives on the use of campus food pantries and student-suggested solutions to encourage use. In response to this gap in the literature, this study serves as a formative phase to explore and intervene in improving college students' use of these resources. The aims of this study were (1) to explore the barriers to using the campus food pantry by college students and (2) to gain insights into student-suggested recommendations to improve food pantry use.

## Materials and methods

### Design and study population

This study was conducted at the University of Florida (UF) in response to growing concern about campus food insecurity and the establishment of a campus food pantry, an emergency food program that launched in the fall of 2016. An exploratory descriptive approach was adopted, given the lack of literature describing the role of campus food pantries and their impact on the food security status of college students. The study employed two recruitment methods: active recruitment of students at the campus food pantry and passive recruitment through distribution of flyers on-campus and e-mail listservs. In qualitative research, the use of a combination of passive and active recruitment techniques has been found to facilitate recruitment [13]. While the general purpose of the study was to recruit food insecure students, the flyer described the study as an investigation of food access, due to the sensitive and potentially stigmatizing nature of the topic of food insecurity.

Students were eligible if they were (1) 18 years of age or older, (2) enrolled as an undergraduate or graduate student at the University of Florida, and (3) answering affirmatively (often true or sometimes true versus never true) to questions 1 and/or 2 of the Adult Food Security Survey Module (AFSSM). These included "within the past 12 months, the food I bought did not last and I did not have money to get more" and "within the past 12 months, I worried whether my food would run out before I got money to buy more". Interested students completed an online screener, and eligible students were invited to participate in semi-structured interviews. Participants completed a written informed consent and received compensation in the form of a $30 gift card. The Institutional Review Board at the University of Florida approved the study protocol prior to recruitment and data collection.

### Data collection

**Semi-structured interviews.**   Individual, face-to-face, semi-structured in-depth interviews (45–60 min in duration) were conducted between September of 2019 and February of 2020 with a purposive sample of 41 undergraduate and graduate college students enrolled at the University of Florida. Eligible participants met with the moderator and a research assistant in a private room on campus. The author AE served as the moderator of all sessions while another team member (research assistant) took written notes. AE is a female PhD candidate who has received training on the conduct of qualitative research [14,15] in addition to a refresher on recognizing the researcher's cognitive biases, pre-conceptions, and internal emotions prior to data collection [16].

AE and the research assistant introduced themselves at the beginning of the interview and described the purpose of the project as a part of the moderator's dissertation work. After providing written informed consent, participants were given an online self-administered survey to complete. The moderator followed a semi-structured interview script that was developed by the research team after a thorough review of the food assistance literature and reviewed by the population of interest and qualified faculty with topic expertise in nutrition, public health, and health communication (see S1 Appendix). The semi-structured interview script included

open-ended questions with probing queries to elicit deeper student responses. The questions explored the student's perceptions of the current campus food pantry, barriers to utilizing the pantry for food acquisition, and suggested solutions. All discussions were audio-recorded with the participants' consent. After each interview, the moderator and research assistant debriefed and discussed the main points, new ideas, and overall impressions.

**Questionnaire.** *Food security status and pantry use.* Food security status over the past 12 months was assessed using the 10-item AFSSM [17]. Participants responded affirmatively or non-affirmatively regarding several dimensions of food insecurity, including anxiety over food supply, meal skipping, diminished quantity and quality of food consumed, and running out of supplies due to a lack of resources to purchase food. Participants were then designated into the following 4 categories: high food security (no food access problems), marginal food security (anxiety over food sufficiency), low food security (adequate amount of food but reduced food quality, variety, or desirability), or very low food security (disrupted eating patterns and reduced food intake). According to the USDA definitions, these categories were further collapsed into either *food secure* (high and marginal food security status) or *food insecure* (low or very low food insecurity status). All scoring procedures were in line with the Guide to Measuring Food Security [18] and the USDA's definitions of food security [2].

To assess food pantry use, participants were asked whether they had ever used the campus food pantry. For those answering affirmatively, they were asked about the frequency of use: "In the semester that you used the campus food pantry the most, how many times did you use it?" Response options were "2–3 times/week; 1 time/week; 1–2 times/month; and less than 1 time/month".

*Sociodemographic questions.* The remaining questions captured demographic and economic variables. Demographics included sex (male/female), age, marital status (married/not married), race (White/Black/Asian/Multi-racial), Hispanic/Latino (yes/no), International student (yes/no), student class (Freshman, Sophomore, Junior, Senior, Graduate) and place of residence. Economic variables included employment (full-time/part-time/not employed), receipt of financial aid (yes/no) and a Pell Grant (yes/no).

## Data analysis

Quantitative data were analyzed using SPSS Statistics for Windows, version 24 (Armonk, NY: IBM Corp). Descriptive statistics were used to summarize sociodemographic characteristics, food security status, and food pantry use. These included frequencies and proportions for categorical variables with standard deviations for continuous variables.

As for the qualitative analysis, audio-recorded interviews were transcribed verbatim and managed using NVivo 12 (QSR, Melbourne, Australia), a qualitative data software package. The author AE analyzed the data independently and iteratively using inductive, semantic thematic analysis by following the steps outlined by Braun and Clarke [19]. Data analysis started by getting familiar with the data and noting down initial ideas. Transcripts were read and re-read by the first author (AE) to identify the initial set of codes. Based on identifying the initial codes, the author AE created a coding scheme and sorted through the different codes to identify potential themes. Emergent themes were also identified and added to the coding structure iteratively when necessary. Ongoing analysis was continued to refine the specifics of each theme and generate clear definitions. After further meetings and discussions with the research team, the coding categories and relevant text were examined again to reduce overlap and redundancy among the categories. The set of candidate themes was reviewed and collapsed into three main themes and eight subthemes with exemplar quotes to illustrate the developed categories.

To increase the rigor of the data collection and analysis, the research team adopted measures to increase credibility and reflexivity. To ensure credibility, the team relied on the audio-recorded and transcribed interviews as the main data repository and used exemplary quotes to support each theme and subtheme. Additionally, to increase reflexivity, the moderator practiced ongoing critical reflection, including personal reflexivity, functional reflexivity, and disciplinary reflexivity on the extent to which her thoughts, actions, and discipline shape the interpretation of data [16]. The research team has also utilized the Consolidated Criteria for Reporting Qualitative research (COREQ) checklist to report study results [20] (See S2 Appendix).

Interviews continued until clear themes arose from the data and discussion of these themes yielded no new information. A theme was saturated when it became redundant and/or replicated across the interviews [21]. Once saturation was attained, the recruitment of participants stopped. To enhance the trustworthiness of the analysis process, authentic quotes from study participants are provided.

## Results

### Characteristics of study participants

Overall, 41 participants completed the study. The majority were women (70.7%) with an average age of 23.7 ± 5.9 years (Table 1). Almost half of the participants classified themselves as White (46.3%), with the remaining identifying as Asian (26.8%), multi-racial (17.1%), and Black (9.8%). Most participants were undergraduate students (70.8%), with the remainder graduate students. About two-thirds were employed (part-time/full-time) and were financial aid recipients. Most of the participants were categorized as food insecure (82.5%), and a third utilized the campus food pantry for food acquisition. Among pantry users, almost half used the food pantry once/week.

**Table 1. Sociodemographic characteristics of student participants (n = 41).**

| Characteristic | No. | % or ± SD |
|---|---|---|
| **Sex** | | |
| Male | 12 | 29.3 |
| Female | 29 | 70.7 |
| **Race** | | |
| White | 19 | 46.3 |
| Black | 4 | 9.8 |
| Asian | 11 | 26.8 |
| Multi-racial | 7 | 17.1 |
| **Hispanic/Latino** | | |
| Yes | 10 | 24.4 |
| No | 31 | 75.6 |
| **Mean age, y** | 23.7 | 5.9 |
| Married | | |
| Yes | 5 | 12.2 |
| No | 36 | 87.8 |
| **Class** | | |
| Freshman | 4 | 9.8 |
| Sophomore | 10 | 24.4 |
| Junior | 6 | 14.6 |
| Senior | 9 | 22.0 |

(*Continued*)

**Table 1.** (Continued)

| Characteristic | No. | % or ± SD |
|---|---|---|
| Graduate Student | 12 | 29.3 |
| **International** | | |
| Yes | 11 | 26.8 |
| No | 30 | 73.2 |
| **Employment** | | |
| Employed part-time | 20 | 48.8 |
| Employed full-time | 6 | 14.6 |
| Unemployed | 15 | 36.6 |
| **Receipt of a Pell Grant** | | |
| Yes | 14 | 34.1 |
| No | 27 | 67.5 |
| **Receipt of Financial Aid** | | |
| Yes | 24 | 58.5 |
| No | 17 | 41.5 |
| **Food Security Status– 2 level coding** | | |
| Food secure | 8 | 19.5 |
| Food insecure | 33 | 82.5 |
| **Food Security Status– 4 level coding** | | |
| High | 1 | 2.4 |
| Marginal | 7 | 17.1 |
| Low | 11 | 26.8 |
| Very low | 22 | 53.7 |
| Ever used campus food pantry | | |
| Yes | 27 | 65.8 |
| No | 14 | 34.2 |
| **Frequency of pantry use[b]** | | |
| 2–3 times/week | 4 | 19.0 |
| 1 time/week | 10 | 47.6 |
| 1–2 times/month | 3 | 14.3 |
| Less than 1 time/month | 4 | 19.0 |

[a]During the 2019–2020 academic year.

[b]Displayed only for pantry users.

## Reasons for not using the campus food pantry

The reasons participants gave for not using the campus food pantry could be summarized under two broad themes. First, respondents were either "choosing not to use the food pantry" or had been unable due to "barriers". Themes and subthemes are presented in Table 2.

## Theme 1: Choosing not to use the campus food pantry

**1.1 Stigma and shame.** Students described the food pantry as a place that induces feelings of "embarrassment" and "shame". These feelings stemmed from the perceived stigma of "taking handouts". Among students with food insecurity, both pantry users and non-users articulated these emotions; however, feelings of stigma were more common among students who did not utilize the food pantry (75.0% of non-pantry users compared to 52.8% of pantry users) (Table 3). As some students expressed:

**Table 2. Themes and subthemes from semi-structured interviews with university students (n = 41).**

| |
|---|
| **Theme: Choosing not to use the campus food pantry** |
| Subtheme: Stigma and shame |
| Subtheme: Perceived insufficient need |
| Subtheme: Unsuitable food |
| **Theme: Barriers** |
| Subtheme: Lack of knowledge |
| Subtheme: Limited food access |
| **Theme: Student-suggested solutions** |
| Subtheme: Spread awareness with positive marketing messages that de-stigmatize use<br> • Categories:<br> ○ Utilize testimonials and demonstrate diversity<br> ○ Utilize inclusive marketing<br> ○ Emphasize the right to food security, prioritize health, and highlight presence of non-canned items<br> ○ Organize events and classroom discussions |
| Subtheme: Improve accessibility of fresh produce and protein options |
| Subtheme: Improve access through satellite locations and online ordering |

> I am a 47-year-old guy with a family. It would be pretty embarrassing if I went in there and one of them [students] saw me. . . I'd feel like I am a substandard father and husband–
> Graduate student, male, non-pantry user with food insecurity

> I might look over my shoulders for that one time that I went to the food pantry. I do not want someone who I had known previously to see me going there–Undergraduate student, female, food insecure, pantry user

Students were concerned about being judged when carrying pantry bags back to class or other campus work sites. For example, one graduate student with food insecurity described feeling anxious when taking pantry grocery bags to her laboratory as it cannot go unnoticed. Students reported being "afraid" of being seen with pantry bags and not wanting anyone to pity them as "using the food pantry would be almost identifying as underprivileged". Another student acknowledged a similar experience:

> When I first started going to the pantry and started coming back with these grocery bags, people would look at me in a weird way–Undergraduate student, male, pantry user with food insecurity

Students also described the competitive spirit at the university as a driver of the stigma around food pantry use. For example, one food insecure non-pantry user explained that "the wonderful competitive spirit here [university] leads to a false sense of not wanting to rely on anyone and try not to be seen as weaker in the fight". Another respondent described her

**Table 3. Frequency of codes discussed as barriers by food pantry use (n = 41).**

| | Pantry User (27) | | Non-User (14) | | Total (41) |
|---|---|---|---|---|---|
| | **Food Secure (6)** | **Food Insecure (21)** | **Food Secure (2)** | **Food Insecure (12)** | |
| **Stigma and shame** | 50.0% | 52.3% | 50.0% | 75.0% | 58.5% |
| **Limited Food Access** | 16.6% | 57.1% | 50.0% | 8.3% | 36.5% |
| **Perceived Insufficient Need** | 16.6% | 28.5% | 50.0% | 50.0% | 34.1% |
| **Lack of Knowledge** | 33.3% | 28.5% | 0% | 50% | 34.1% |
| **Unsuitable food** | 0% | 33.3% | 0% | 0% | 17.0% |

generation as "prideful", which makes it difficult to ask for help, especially when her friends are "OK spending $15 on a pizza". Therefore, the idea of using the pantry made students feel self-insufficient in comparison to their peers.

**1.2 Perceived insufficient need.** Another common barrier was the perception that obtaining food from the pantry would take resources from others who need it more. Many students with food insecurity, despite need, identified as undeserving of this resource. Participants expressed concerns over "taking away from someone's portion or food". The sense that others are in greater need deterred them from using the pantry. As quoted:

> I thought that it could alleviate some cost that I have from food but I don't want to be taking from someone else–Undergraduate student, female, non-pantry user with food insecurity

Although common among all participants, non-pantry users with food insecurity reported this barrier more frequently (50.0% vs. 28.5% of pantry users with food insecurity). Non-users distanced themselves from using the food pantry because it was perceived as a resource only for "financially-deprived" people. Indeed, when discussing the intended clientele for food pantries, respondents described vulnerable populations like those who need "a lot of financial aid", "students down to the wire", those with "desperate need", "in crisis" or "on social benefits". For example, one student stated:

> If I can work 20 hours a week, then why should I use the food pantry? It should be used by those who are on social benefits or using those SNAP or ability cards–Undergraduate, female, non-pantry user with food insecurity

However, no respondent assessed their "need" to use the food pantry services based on food security standards. In fact, many students were not aware of what food insecurity meant and viewed it as a normal part of the college experience. As quoted:

> A lot of students don't even know, technically, what food insecurity means. If they knew what it actually meant, I think they would consider themselves food insecure–Graduate student, female, pantry user with food insecurity

> This is just how colleges are. It is all about starving yourself and being broke–Undergraduate student, female, pantry user with food insecurity

Using the food pantry was deemed as a "last resort" for some students. These students did not necessarily reflect a lack of food insecurity, but rather an absence of a severe need to use the food pantry, underlying subjective understandings of the concept of "need" and of those for whom assistance is intended. For example, one international student described her challenges with ensuring adequate food supply; however, when asked about using the food pantry for assistance, she stated:

> Even though my economic situation is not great because of my international status and my parents being in Venezuela, I feel like there are people worse than I am. There are people who need it more so that's one of the reasons I haven't considered going–Undergraduate student, female, non-pantry user with food insecurity

Campus marketing messages appear to influence students' perceptions of who the pantry is intended to help. Many non-pantry users with food insecurity reported that introducing the food pantry as "a place for low-income people is really harmful". Marketing materials using

other terms like "financially-deprived people" or those who "go hungry" also deterred students from seeking help. As a result of these marketing terms, students felt the moral imperative of not taking advantage of such a resource. As some students expressed:

> I think in one of the posters I have seen it written that it's like "ten cents a can for financially deprived people" or something similar. So, there's always this ego that "I can earn this for myself. I don't need this to access food"–Graduate student, male, pantry non-user with food insecurity

> There has to be a better way to introduce the pantry. . .as soon as you start mentioning, 'if you go hungry all the time', people get afraid to use it–Undergraduate student, female, pantry user with food insecurity

**1.3 Unsuitable food.**   Students described dissatisfaction with the food provided in the food pantry and reported that the quality of foods offered made it not worthwhile to go there. This theme was most reported by pantry users with food insecurity. Respondents highlighted the lack of fresh foods (specifically fruits and vegetables), "healthy foods", or high protein foods. Students also described frequently receiving instant foods like macaroni and cheese, "tinned foods", and foods that were past their "best by" dates. Thus, perceptions of food type and quality dissuaded respondents from asking for help. The following quotes illustrate the nature of responses in this category:

> It's all like frozen stuff, and I am more concerned about the expiration date. There were a couple of boxes with the best before date had passed. I always check because I am going to give that food to my kid–Graduate student, female, pantry user with food insecurity

> There is a lot of canned food and I don't like the artificial flavor of the tinned food–Graduate student, female, pantry user with food insecurity

## Theme 2: Barriers

**2.1 Lack of knowledge.**   Half of non-pantry users with food insecurity indicated that they lacked knowledge on how to use the food pantry. Students were unfamiliar with the location and hours of operation. They also questioned their eligibility to use the pantry, despite it being open to anyone with a university ID. Students also assumed there were "all these hoops" and paperwork required before using the food pantry. As quoted:

> Two to three years I walked by that building [food pantry], every day, and I didn't realize you could just walk in and get food if you need it. I thought it has something to do with the meal plans that cost money–Undergraduate student, female, pantry user with food insecurity

> I am still not clear who can use the food pantry because many people think it is just for financially deprived people, but most students are struggling–Undergraduate student, male, non-pantry user with food insecurity

> I think the biggest barrier is knowledge, the knowledge that it's there and what it does, and what it expects of you–Undergraduate student, female, pantry user, food secure

Students reported that the pantry is rarely advertised on campus and recommended spreading information through on-campus promotional messages. Other students suggested the need to bring awareness to the issue of food insecurity as a whole:

I know a lot of students don't even know about it [food pantry] in the first place so more awareness in general would help–Undergraduate student, female, pantry user with food insecurity

It does not look like they advertise it [food pantry] a lot. And even not a lot of students know about it–Graduate student, male, pantry user with food insecurity

**2.2 Limited food access.** About 37% of the overall sample and 57% of food pantry users with food insecurity indicated that they had attempted to use the food pantry in the past, but they did not find it feasible. Some of the barriers included the lack of availability of desirable food, the potential for "missing out" on fresh produce, and long lines:

I go on Mondays because as soon as they open they will run out of food. Like yesterday I went but I didn't take anything because there was only green beans left and the entire refrigerator was empty–Graduate student, female, pantry user with food insecurity

Other pantry users with food insecurity reported being disappointed in the fruit and vegetable section:

For the first two years, I loved it. They had a lot of fruits and veggies and they had kale leaves and all. But now I don't see any of those things–Graduate student, female, pantry user with food insecurity

Some respondents also described distaste at feeling like they had to "fight for access" which "dampened the motivation to come back". The economic cost of their time in long lines, combined with the feeling of having to compete with other pantry users, deterred their food pantry use. This was described by one student:

It's so competitive to get anything in there; people were pushing people out of the way trying to get to the produce section. . . so that was kind of like a crazy experience. I was like, "wow, I didn't understand it was like this competitive to get food"–Undergraduate, female, pantry user with food insecurity

### Theme 3: Student-suggested solutions

Students suggested three main solutions to addressing barriers to utilizing the campus food pantry. These included (i) spreading awareness with positive marketing messages that de-stigmatize use, (ii) improving accessibility of fresh produce and protein options, and (iii) improving access through satellite locations and online ordering systems.

**3.1 Spread awareness with positive marketing messages that de-stigmatize use.** To overcome the lack of information and hesitance in using the pantry, students suggested potential solutions. Suggestions included using testimonials from diverse members of the student body, implementing inclusive messaging, and emphasizing students' right to food and nutrition when spreading awareness on-campus.

*3.1.1 Utilize testimonials and demonstrate diversity.* Respondents recommended the use of stories of people from different backgrounds who have used the pantry and benefited from its food products. Students felt that sharing stories of how the food pantry has helped others would decrease the social stigma associated with its use:

Have more stories of people using it with different backgrounds and show that they do not have shame in going–Undergraduate student, female, non-pantry user, food secure

I understand the social stigma surrounding the food pantry, but to squash those issues I think you have to tell everybody what the food pantry does for a lot of people"–Graduate student, female, pantry user with food insecurity.

*3.1.2 Utilize inclusive marketing.* One of the most common solutions suggested by respondents was advertising the food pantry as a "no-judgment zone" and "a place for everyone". Respondents recommended that marketing should make others feel that "it is not just you; a lot of people need this" and should speak to food insecure students who may not identify as food insecure. For example, one student stated:

Show it like a safe zone. There's no judgment, and you know it's a fun thing to do to grab groceries in between classes–Undergraduate student, male, pantry user with food insecurity

Explain to students that, "hey, there's no shame in this game. . .there is nothing you can do when you are broke, but you can do this, and no one's going to judge you for it [using the pantry]" Undergraduate student, female, pantry user with food insecurity

There were also suggestions about refraining from using the term "in need" due to its vague definition and "association with poverty":

When it's described, say, "Hey, this isn't a poor person's handout. This is just a little pick-me-up". Use all these nice euphemisms that don't offend people when they feel ashamed or guilty about it–Undergraduate student, female, non-pantry user with food insecurity

*3.1.3. Emphasize the right to food security, prioritize health, and highlight presence of non-canned items.* Respondents recommended shifting marketing messages from portraying the pantry as a charitable resource to prioritizing health and emphasizing the right to nutrition. This type of marketing is likely to decrease perceptions of stigma and help normalize its use among students:

Maybe prioritize health? As part of a marketing campaign? Because many people say they don't eat healthy, so you could introduce it that way. And you can mention how people can't afford to be healthy and that they offer healthy options there [food pantry]. I think that would motivate students to use it–Undergraduate student, male, pantry user with food insecurity

Additionally, shedding light on the availability of fresh produce and protein-rich foods, and "anything but canned foods" would be an added benefit:

Don't mention green beans! . . . part of the health campaign, mention that, that they won't just give you canned food–Undergraduate student, female, pantry user with food insecurity

*3.1.4 Organize events and classroom discussions.* Respondents suggested advertising for the food pantry during school orientation sessions as well as organizing events "to make it look less like a food bank and more attractive". For example, one student suggested having "events at the pantry so people can get accustomed to using it". These events can be in the form of cooking classes, "farmer's market booth once a month" or tabling events with recipes that can be made with food products from the campus pantry:

Cooking classes in the pantry are a really good idea. I feel like you could use a lot of the food that's already at the pantry to make recipes; that way they can go and get that same food products and make those foods that are nutritious so that they're not just eating ramen–Undergraduate student, female, pantry user with food insecurity

If they could just have an open house. Just something to let people know that it is there, some kind of event to make people feel like they are not weird or poor–Undergraduate student, female, non-pantry user with food insecurity

In addition to organizing events, students suggested that instructors should highlight the food pantry when discussing syllabi and resources on campus "because if nobody talks about it, students are not going to know about it; they're not going to go get food".

**3.2 Improve accessibility of fresh produce and protein options.**   To encourage food pantry use by students with food insecurity and help students achieve a balanced diet, students suggested adding more fruits and vegetables as these are the type of foods that they can rarely afford:

The only thing I would add is fruits and vegetables. I would like to see more of that–Undergraduate student, female, pantry user with food insecurity

Respondents also recommended having more protein options, including a frozen meat section:

I did see tuna there. I think that it was a good idea. So having more protein options as well, that would be really, really, helpful for a lot of students–Undergraduate student, male, pantry user with food insecurity

**3.3 Improve access through satellite locations and online ordering.**   Finally, implementing an online ordering system where students can place an order ahead of time and pick up their basket later would aid in alleviating stigma and decreasing scheduling conflicts. Since the pantry staff started using this system during the COVID-19 pandemic, it could be continued even after return to normal operations. Additionally, having small satellite locations in residence halls may further improve access:

One thing that would really help is a grocery pickup system like Walmart where you can order online and pickup later. It could really help students with time and scheduling–Undergraduate student, male, pantry user, food secure

Having remote locations for pick-up or online orders will make students want to use it more because they're going to feel less embarrassed–Undergraduate student, male, non-pantry user with food insecurity

## Discussion

This qualitative study provides practical and important insights on factors preventing college students from accessing campus food assistance and highlights students' recommendations for overcoming these barriers. Although many students may need campus food assistance, food pantries may not be reaching food insecure students due to reasons beyond resource availability. Our findings illustrate how an understanding of certain perceptions could help colleges encourage more food insecure students to use campus food pantries when available.

Respondents described physical access barriers like the unavailability of specific foods and a lack of knowledge regarding the pantry's "how to" procedures. Moreover, these barriers were intertwined with perceptions pertaining to social stigma, perceived insufficient need, and availability of only "canned food" that, in some instances, stemmed from specific marketing messages. A modest portion of students expressed their gratitude and thankfulness for the existence of such a resource on campus; however, these students emphasized the need to improve the accessibility of fresh produce and protein options, coupled with positive marketing that destigmatizes use. Adding remote locations or an online ordering system may further normalize the shopping experience, decrease long lines, and alleviate embarrassment.

Feelings of stigma and shame remain the main deterrent to using campus food pantries among students with food insecurity, a finding that has been corroborated by previous studies [10,22]. These feelings of embarrassment were often accompanied by the perception that others may have greater needs and feelings of guilt about "taking advantage" of the food pantry. Both shame and perception of insufficient need may relate to the desire for control and independence. Some food insecure students, however, were able to overcome feelings of stigma by viewing their situation as one requiring "temporary help" or by putting pantry use in the context of previous experience with using food assistance with their families.

Efforts to normalize pantry use by portraying it as a "no-judgment, guilt-free zone" are needed. These can be accomplished through anti-stigma campaigns that show students that they are "not alone" in needing food assistance. Further, campus pantries can be a site for nutrition literacy events that shift the pantry's image from a charity-based program to a hub for nutrition resources. These events can include small-group nutrition education programs, hands-on cooking classes for creating healthy meals on a budget, and "farmers'-market-style" tabling events where students have quick access to fresh produce grown locally at the campus gardens.

Many food insecure non-pantry users who could have benefited from the pantry services noted that earlier marketing messages have had unintended consequences, a theme that aligned with previous reports of the underlying reasons behind the perception of the insufficient need barrier [23]. Students reported that constantly advertising the pantry as a place for low-income individuals is "harmful" because it portrayed users as needy or victims, violating the shared inheritance of self-reliant individuality and deterring students from seeking help. Additionally, the use of the phrase "in need" has often left students confused as to who the pantry was intended to benefit.

To encourage students with food insecurity to seek help, campus messages should utilize gain-framed messages to highlight the pantry's resources and prioritizing health. Gain-framed messages are suggested to be effective when targeting prevention behavior [24,25], such as the prevention of food insecurity and hunger. For example, instead of portraying the pantry as a place for the "financially-deprived", promotions should rebrand the pantry as a community resource that students can use to obtain a healthy, balanced diet. Highlighting resourcefulness rather than dependence may also assist in reducing feelings of shame and promoting more social inclusion of students seeking assistance.

The perception of the pantry offering poor quality food–with some canned goods being past the "best by" date–led to a distaste for seeking assistance from the food pantry and influenced the students' subjective understandings of social status and self-worth. In addition to ensuring that pantry users do not receive foods past their "best by" date, universities should seek input from students about the types of foods that are most desirable (such as fresh produce and animal/plant-based proteins as shown in this study). Seeking such input can serve multiple goals: 1) building long-term community partnerships and enabling interested donors to donate appropriate foods, 2) reducing food waste and, 3) ensuring that students maintain

their sense of self-worth and respect from service providers. Lastly, pantry administrators could consider providing recipes specific to shelf-stable foods to increase perception of acceptability and usefulness by pantry users.

Observation of long lines, crowds, and empty shelves dissuaded potential users with food insecurity from re-visiting the food pantry. To overcome this, campus food pantry administrators can aim to distribute supplies evenly throughout the week to ensure a more consistent supply. Additionally, having an online ordering system in place can assist with limiting crowds and reducing the "fight for food" perception compared to an in-person shopping experience. Through this system, students also could be notified online about the availability of food supplies to avoid disappointment with limited food options.

This study has some limitations that need to be acknowledged. First, although moderators underwent training, there is always a possibility of interviewer bias that has the potential of influencing the depth of participant discussion. Second, the qualitative nature of this study and face-to-face interviewing may have introduced social desirability bias, prompting fewer responses that show dependence on a non-profit service. Third, the study explored the experiences of students at one institution–a large public university in the Southeast. Thus, the results cannot be generalized to other institutions, especially those found at private universities or at colleges serving primarily nontraditional student populations. Nevertheless, this was the first study to qualitatively explore barriers to utilizing a campus food pantry and provide student-suggested solutions to overcoming these barriers, thus significantly contributing to a growing body of research on food insecurity and food assistance use nationwide. Conducting interviews with diverse student groups, including more racial minorities and graduate students with families may offer additional findings. Future research also should examine different food pantry models and assess the effectiveness of interventions aiming to reduce stigma and negative perceptions associated with food assistance.

## Conclusion

This is one of the first studies to examine qualitatively students' experiences with on-campus food pantry use, and it highlights the presence of access and resistance obstacles. The barriers identified and proposed solutions need to be systematically addressed in future food assistance interventions for college students with food insecurity. Progress can be made through working with students by portraying the food pantry as a hub for food literacy activities, providing desirable food, evenly distributing resources, and communicating through positive marketing messages. These marketing messages should serve to increase knowledge of existing resources and de-stigmatize seeking assistance by emphasizing resourcefulness rather than "neediness". While universities cannot rely on a traditional food charity model to address student food insecurity, the development and testing of different models for campus food pantry operations, improving its resource availability, and implementing the right messaging can provide a partial remedy.

## Supporting information

**S1 Appendix. Topic guide for semi-structured interviews with university students.** (DOCX)

**S2 Appendix. Consolidated criteria for Reporting Qualitative research (COREQ) checklist.** (DOCX)

## Acknowledgments

The authors wish to thank all participating students for their time and contributions to this study.

## Author Contributions

**Conceptualization:** Aseel El Zein, Melissa J. Vilaro, Karla P. Shelnutt, Anne E. Mathews.

**Data curation:** Aseel El Zein.

**Formal analysis:** Aseel El Zein, Melissa J. Vilaro.

**Funding acquisition:** Aseel El Zein, Anne E. Mathews.

**Investigation:** Aseel El Zein, Melissa J. Vilaro, Anne E. Mathews.

**Methodology:** Aseel El Zein, Melissa J. Vilaro, Karla P. Shelnutt, Kim Walsh-Childers, Anne E. Mathews.

**Project administration:** Aseel El Zein, Anne E. Mathews.

**Resources:** Aseel El Zein, Melissa J. Vilaro, Kim Walsh-Childers, Anne E. Mathews.

**Software:** Aseel El Zein.

**Supervision:** Aseel El Zein, Melissa J. Vilaro, Anne E. Mathews.

**Validation:** Aseel El Zein, Melissa J. Vilaro, Anne E. Mathews.

**Visualization:** Aseel El Zein, Anne E. Mathews.

**Writing – original draft:** Aseel El Zein.

**Writing – review & editing:** Aseel El Zein, Melissa J. Vilaro, Karla P. Shelnutt, Kim Walsh-Childers, Anne E. Mathews.

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
