## [Decision Letter · Decision Letter 0]

11 Jan 2022

PONE-D-21-32850OBSTACLES TO UNIVERSITY FOOD PANTRY USE AND STUDENT-SUGGESTED SOLUTIONS: A QUALITATIVE STUDYPLOS ONE

Dear Dr. Mathews,

Thank you for submitting your manuscript to PLOS ONE. After careful consideration, we feel that it has merit but does not fully meet PLOS ONE’s publication criteria as it currently stands. Therefore, we invite you to submit a revised version of the manuscript that addresses the points raised during the review process. In particular please address comments made by reviewer 1.

We look forward to receiving your revised manuscript.

Kind regards,

Catherine Haighton, PhD

Academic Editor

PLOS ONE

Journal Requirements:

(No authors have competing interests.)

Reviewers' comments:

Reviewer's Responses to Questions

**Comments to the Author**

1. Is the manuscript technically sound, and do the data support the conclusions?

Reviewer #1: Partly

Reviewer #2: Yes

2. Has the statistical analysis been performed appropriately and rigorously? 

Reviewer #1: Yes

Reviewer #2: N/A

3. Have the authors made all data underlying the findings in their manuscript fully available?

Reviewer #1: No

Reviewer #2: Yes

4. Is the manuscript presented in an intelligible fashion and written in standard English?

Reviewer #1: Yes

Reviewer #2: Yes

5. Review Comments to the Author

Reviewer #1: - There is a lack of consideration for the relationship between researcher/s and participants. Were all 3 researchers present at the interviews? If so, this brings a possible power balance and intimidation that may affect responses, which was not considered within the limitations. Was a relationship / rapport built with participants prior to interviewing? 3 interviewers would be an even larger issue if not, if so this should be reported.

-It would be beneficial to see the interview schedule, and how it changed as a result of feedback from public health / nutrition professionals and a trial run with participants (what were the 'edits' you employed?).

-One issue is that although the authors are noted to have attended training on the ethical conduct of research including researcher bias, these potential biases are not specified.

-It would be useful to see more consistent description of the analysis approach used for the interviews. In the abstract (which is very clear and thorough) you describe content analysis, but within the methodology this terminology is not used, and instead is described as a mixture of inductive and deductive (line 187). More clear reporting on the theoretical background (if any) of the analysis approach, and consistency across abstract and methodology, is needed.

-Where are the field notes? And data for the background information of the sample?

Reviewer #2: An interesting topic and one which is very timely.

Abstract is clear and concise.

Glad to see limitations clearly discussed. On the whole a fitting paper for this journal and with some minor revisions, I would accept in its current form.

6. PLOS authors have the option to publish the peer review history of their article (what does this mean?). If published, this will include your full peer review and any attached files.

Reviewer #1: **Yes: **Bethany Nichol

Reviewer #2: No

---

## [Author Response · Author response to Decision Letter 0]

8 Mar 2022

Dear Editor:

We would like to thank our reviewers for a thoughtful critique of our manuscript, “Obstacles to University Food Pantry Use: A Qualitative Study”. Below, we have addressed reviewer’s 1 critique, referencing changes to our manuscript where applicable. We hope these revisions are received favorably by reviewers and the editorial staff.

We have updated the format of the tables, authors’ affiliation, and manuscript body. 

(No authors have competing interests.)

The authors have declared that no competing interests exist. 

We would like to change our Data Availability Statement to the following:

Data cannot be shared publicly because of confidentiality and privacy concerns. The University of Florida Institutional Review Board approved the study, and the consent document the IRB approved assured participants that their data would not be shared beyond the research team and as aggregated in publications. 

Captions for supporting information were added to the end of the manuscript and in-text citations were updated accordingly. 

6. Please review your reference list to ensure that it is complete and correct. If you have cited papers that have been retracted, please include the rationale for doing so in the manuscript text, or remove these references and replace them with relevant current references. Any changes to the reference list should be mentioned in the rebuttal letter that accompanies your revised manuscript. If you need to cite a retracted article, indicate the article’s retracted status in the References list and also include a citation and full reference for the retraction notice

Reference list was updated. 

Response to reviewers

Reviewer #1: - There is a lack of consideration for the relationship between researcher/s and participants. Were all 3 researchers present at the interviews? If so, this brings a possible power balance and intimidation that may affect responses, which was not considered within the limitations. Was a relationship / rapport built with participants prior to interviewing? 3 interviewers would be an even larger issue if not, if so this should be reported.

Thank you for the opportunity to add further clarity. Two researchers were present mostly at all times. The researcher (AE) served as the moderator for the interviews while one of the two research assistants acted as an observer and note-taker. This is now clarified on page 7, lines 147-149.

Relationship with participants: 

There was no pre-interview activity or contact between the participants and interviewers. Participants were not given information about the interviewer beyond a brief introductory statement at the beginning of the interviews that described the interviewer’s role in the study and status (PhD candidate at the University of Florida).

This is now clarified on page 7, lines 153-155. The protocol was IRB approved and standard protocol for social and behavioral research that are acceptable to participants were used.

We have also uploaded a COREQ checklist for qualitative research in supporting materials (See S2 Appendix). 

-It would be beneficial to see the interview schedule, and how it changed as a result of feedback from public health / nutrition professionals and a trial run with participants (what were the 'edits' you employed?).

The interview schedule was uploaded (See S1 Appendix). 

We have also decided to take out the following sentence: “To improve the conversational flow and ensure clarity, applicability, and feasibility, the interview guide was tested with the population of interest” upon reflection and considering the reviewer comments regarding the edits that were made. The changes were minor overall (minimal language/grammar changes). 

We have added the “population of interest” to the following sentence “The moderator followed a semi-structured interview script that was developed by the research team after a thorough review of the food security and food assistance literature and reviewed by the population of interest and qualified faculty with topic expertise in nutrition, public health, and health communication”. We feel this more accurately reflects the process without implying to the reader that the changes based on the student review were such that they would denote more expansive explanation.

-One issue is that although the authors are noted to have attended training on the ethical conduct of research including researcher bias, these potential biases are not specified.

Thank you for your comment. We have clarified the sentence and cited the trainings, page 7, lines 150-152. 

-It would be useful to see more consistent description of the analysis approach used for the interviews. In the abstract (which is very clear and thorough) you describe content analysis, but within the methodology this terminology is not used, and instead is described as a mixture of inductive and deductive (line 187). More clear reporting on the theoretical background (if any) of the analysis approach, and consistency across abstract and methodology, is needed.

Thank you for pointing this out. We misnamed the approach used in the previous version of the manuscript. Edits were implemented to describe the inductive, semantic thematic analysis used to guide data analysis. This is now described on page 9, line 205. The abstract was also updated to reflect the same approach.

-Where are the field notes? And data for the background information of the sample?

Data on the background information of the sample can be found in the characteristics of study participants, page 11, lines 234-241 and table 1. This information was also added to the COREQ checklist. 

In health-related social science, we take interview notes which are usually collected by the second researcher present (research assistant in our study). These are reported on page 7, lines 148-149.

---

## [Decision Letter · Decision Letter 1]

7 Apr 2022

Obstacles to university food pantry use and student-suggested solutions: A qualitative study

PONE-D-21-32850R1

Dear Dr. Mathews,

We’re pleased to inform you that your manuscript has been judged scientifically suitable for publication and will be formally accepted for publication once it meets all outstanding technical requirements.

Kind regards,

Catherine Haighton, PhD

Academic Editor

PLOS ONE

Additional Editor Comments (optional):

Reviewers' comments:

Reviewer's Responses to Questions

**Comments to the Author**

1. If the authors have adequately addressed your comments raised in a previous round of review and you feel that this manuscript is now acceptable for publication, you may indicate that here to bypass the “Comments to the Author” section, enter your conflict of interest statement in the “Confidential to Editor” section, and submit your "Accept" recommendation.

Reviewer #1: All comments have been addressed

2. Is the manuscript technically sound, and do the data support the conclusions?

Reviewer #1: Yes

3. Has the statistical analysis been performed appropriately and rigorously? 

Reviewer #1: Yes

4. Have the authors made all data underlying the findings in their manuscript fully available?

Reviewer #1: Yes

5. Is the manuscript presented in an intelligible fashion and written in standard English?

Reviewer #1: Yes

6. Review Comments to the Author

Reviewer #1: Thank you for responding to every one of my comments. I feel that each one has been more than sufficiently addressed and have no further suggested amendments. I appreciate the interview schedule and COREQ checklist as supplementary material to increase transparency too.

7. PLOS authors have the option to publish the peer review history of their article (what does this mean?). If published, this will include your full peer review and any attached files.

Reviewer #1: **Yes: **Beth Nichol

---

## [Editor Report · Acceptance letter]

13 May 2022

PONE-D-21-32850R1 

Obstacles to university food pantry use and student-suggested solutions: A qualitative study 

Dear Dr. Mathews:

I'm pleased to inform you that your manuscript has been deemed suitable for publication in PLOS ONE. Congratulations! Your manuscript is now with our production department. 

Kind regards, 

on behalf of

Dr. Catherine Haighton 

Academic Editor

PLOS ONE